# Antenatal and postpartum prevention of Rh alloimmunization: A systematic review and GRADE analysis

Candyce Hamel[1]*, Leila Esmaeilisaraji[1], Micere Thuku[1], Alan Michaud[1], Lindsey Sikora[2], Karen Fung-Kee-Fung[3,4]

1 Ottawa Hospital Research Institute, Ottawa, Ontario, Canada, 2 Health Sciences Library, University of Ottawa, Ottawa, Ontario, Canada, 3 Department of Obstetrics and Gynecology, Division of Maternal-Fetal Medicine, Ottawa Hospital, Ottawa, Ontario, Canada, 4 Division of Maternal-Fetal Medicine, Faculty of Medicine, University of Ottawa, Ottawa, Ontario, Canada

* cahamel@ohri.ca

**Data Availability Statement:** All relevant data are within the manuscript and its Supporting Information files.

## Abstract

### Background

Existing systematic reviews of Rh immunoprophylaxis include only data from randomized controlled trials, have dated searches, and some do not report on all domains of risk of bias or evaluate the certainty of the evidence. Our objective was to perform an updated review, by including new trials, any comparative observational studies, and assessing the certainty of the evidence using the GRADE framework.

### Methods

We searched MEDLINE, Embase and the Cochrane Library from 2000 to November 26, 2019. Relevant websites and bibliographies of systematic reviews and guidelines were searched for studies published before 2000. Outcomes of interest were sensitization and adverse events. Risk of bias was evaluated with the Cochrane tool and ROBINS-I. The certainty of the evidence was performed using the GRADE framework.

### Results

Thirteen randomized trials and eight comparative cohort studies were identified, evaluating 12 comparisons. Although there is some evidence of beneficial treatment effects (e.g., at 6-months postpartum, fewer women who received RhIg at delivery compared to no RhIg became sensitized [70 fewer sensitized women per 1,000 (95%CI: 67 to 71 fewer); $I^2$ = 73%]), due to very low certainty of the evidence, the magnitude of the treatment effect may be overestimated. The certainty of the evidence was very low for most outcomes often due to high risk of bias (e.g., randomization method, allocation concealment, selective reporting) and imprecision (i.e., few events and small sample sizes). There is limited evidence on prophylaxis for invasive fetal procedures (e.g. amniocentesis) in the comparative literature, and few studies reported adverse events.

**Funding:** The original systematic review RCT portion did not receive specific funding. The observational studies portion and RCT update of this review was funded by the Strategy for Patient Oriented Research (SPOR) Evidence Alliance (https://sporevidencealliance.ca/). The funders had no role in study design, data collection and analysis, decision to publish, or preparation of the manuscript.

**Competing interests:** The authors have declared that no competing interests exist.

## Conclusion

Serious risk of bias and low to very low certainty of the evidence is found in existing RCTs and comparative observational studies addressing optimal effectiveness of Rh immunoprophylaxis. Guideline development committees should exercise caution when assessing the strength of the recommendations that inform and influence clinical practice in this area.

## Introduction

Rhesus D (Rh D) alloimmunization leading to hemolytic disease in the fetus and newborn, a preventable condition, carries a global burden of infirmity, resulting in some 50,000 fetal deaths annually, primarily in low and middle income countries in Asia and Sub-Saharan Africa [1]. In the developed world, however, the introduction of Rh immunoprophylaxis (RhIg) in the 1960s into routine obstetrical care for Rh negative women at risk lead to a dramatic fall in the number of Rh affected babies and is considered an immunological success story in the conquest of hemolytic disease of the fetus and newborn in these countries [2]. Initial clinical trials of postpartum administration of RhIg by Freda and Gorman in the US demonstrated that sensitization to the Rh-D antigen of Rh negative women delivering Rh positive newborns could be reduced more than tenfold from 14% to approximately 1% when susceptible women were administered RhIg after delivery of these offspring [3]. Further gains in prevention were possible when RhIg was also administered to Rh negative women in the antenatal period, as evidenced by Bowman et al, in the Canadian Rh Prophylaxis trials of the 1970s, reducing the incidence of Rh sensitization from 1.8% to 0.07% [4]. These trials, and those like them, informed the development of clinical practice guidelines for prevention of Rh alloimmunization in the developed world, decreasing the burden of a disease previously thought unpreventable, for thousands of women.

Clinical practice guidelines fulfill an important role in establishing a framework for the evaluation of healthcare quality. Modern guideline development relies heavily on evidence gleaned from well conducted systematic reviews (SRs) that not only objectively evaluate the existing research evidence for effectiveness, but also provide direction as to the confidence that can be placed in any recommendations that are made by the guideline development team from this information. The latter function is fulfilled by use of the Grading of Recommendations Assessment, Development and Evaluation (GRADE) methodology. This framework has been adopted by the WHO and internationally by hundreds of organizations as the standard and replaces the previous approach of making recommendations based on levels of evidence [5].

To date, the published SR evidence supporting established protocols for prevention of Rh alloimmunization have been limited by methodological flaws (e.g., suboptimal risk of bias assessment), have been restricted to randomized controlled trials (RCTs), and have not all evaluated the certainty of the evidence [6–8]. As it is not always feasible to conduct RCTs (e.g., cost constraints), data from comparative observational studies may also provide additional evidence. Further, existing guidelines differ in recommendations on timing and dose of RhIg administration [9], use evidence from older SRs, and do not include a rating of the certainty of evidence [e.g., GRADE] beyond rating the evidence on study design [10–15]. As SRs are considered to be essential to produce trustworthy guidelines [16], our objective was to systematically review the evidence from RCT and comparative observational studies, for the effectiveness of RhIg in Rh-negative pregnant and postpartum women at risk of Rh alloimmunization. From this review, we aimed to answer the following primary question: *What is the optimal*

*strategy to administer RhIg for immunoprophylaxis, what is the optimal dose of RhIg in Rh-nega-tive pregnant and postpartum women at risk of Rh alloimmunization, and what is the certainty of the evidence*? A secondary question is: *For what obstetrical conditions is there evidence from RCTs and comparative observational studies to support antenatal immunoprophylaxis*?

## Materials and methods

This review was prepared according to the Preferred Reporting Items for Systematic Reviews and Meta-Analyses (PRISMA) statement [17]. For additional quality control, we used A Mea-surement Tool to Assess systematic Reviews (AMSTAR 2) to guide the conduct of this review [18]. The protocols were registered in PROSPERO prior to starting the reviews (RCT CRD# 42019139610; Observational CRD# 42020161798).

### Eligibility criteria

Eligibility criteria are presented in Table 1.

### Literature search

A search strategy was developed by a health sciences librarian in collaboration with the review team. The search strategy was developed in Medline, and then translated into the other data-bases, as appropriate (S1 File), and peer-reviewed [19]. The finalized trials search was used to create the observational study search (S1 File), with additional lines to identify the study designs (e.g., lines 25–36 in Medline search).

**Table 1. PICOTS.**

| PICOTS | Inclusion criteria | Exclusion criteria |
|---|---|---|
| **Population** | Rh-negative pregnant or post-partum (up to 6 weeks) women | Women who have Rh sensitization from previous pregnancies. |
| **Intervention** | RhIg at any time during pregnancy or in the postpartum period (up to 72 hours postpartum) | |
| **Comparator** | • No RhIg, including placebo<br>• RhIg at another time (e.g. different week of pregnancy, antenatal, postpartum)<br>• Different dosage<br>• Different route of administration (e.g., intramuscularly, intravenously) | |
| **Outcomes** | 1. Sensitization/Rh alloimmunization during pregnancy | |
| | 2. Sensitization/Rh alloimmunization after childbirth | |
| | 3. Maternal adverse events | |
| **Timing** | No time limits. Note: Studies published in 2000 onwards were identified through databases, grey literature, and supplemental searching (e.g., bibliography searching of systematic reviews). Studies published prior to 2000 were identified through grey literature and supplemental searching. | |
| **Study design** | Randomized controlled trials, non-randomized controlled trials (i.e., quasi-randomized), comparative observational studies (i.e., comparative cohort and case-control) | Systematic reviews, narrative reviews, single-arm cohort, case-series, case-reports, cross-sectionals, editorials, letters, commentaries, abstracts |
| **Geographic Location** | Any country | |
| **Setting** | Hospitals, specialty care clinics (e.g., Ob/gyn), birthing centres, primary care (e.g., general practitioner nurses), midwives. | |
| **Language** | English and French* | All other languages. |

* Search strategies were not restricted by language, but those published in languages other than English or French were excluded during full-text screening

Medline and Medline in Process via Ovid and Embase Classic + Embase via Ovid were searched using both the trials and observational studies search strategies, using study design filters. For trials, the Cochrane Central Register of Controlled Trials, Cochrane Database of Systematic Reviews, and Database of Abstracts of Reviews of Effects via Ovid were also searched. As this SR was conducted as one component of informing an update of the published Canadian Clinical Practice Guideline from the Society of Obstetricians and Gynaecologists of Canada (SOGC) [10], which searched from 1968–2001, all databases were searched from January 2000 to November 26, 2019. There were no language exclusion criteria in the searches. Studies published prior to 2000 were captured from the grey literature of national obstetrics and gynecology specialty societies and luminary specialty journals and bibliographic searching. Additional details of the search (e.g., dates, websites, study design filters) can be found in S2 File.

## Study selection

Database results were entered into separate Endnote files (i.e., one for trials and one for observational studies) for processing and to remove duplicates. The remaining unique articles were uploaded into an online systematic review managing software (DistillerSR©). Study selection was performed in two stages. First, two reviewers (CH, LE, MT) screened citations based on titles and abstracts, using the liberal accelerated method [20]. Those considered potentially relevant were further reviewed using the full-text publication against the *a priori* defined inclusion criteria by two independent reviewers (CH, LE, MT, AM), in duplicate. All disagreements were resolved through discussion between the two reviewers with the conflicted record. Both stages of review started with a pilot exercise to ensure consistent application of the screening criteria (S3 File). PRISMA flow diagrams [17] were prepared to document the process of study selection through each stage of review for both RCTs and comparative observational studies.

## Data extraction

Data extraction using DistillerSR, was performed by one reviewer and verified by a second (CH, LE, MT), with conflicts resolved through discussion. Information related to publication characteristics (e.g. authors, year of publication), study population (e.g., age), interventions and comparators (e.g., dosage, timing of administration), outcomes, and study design were gathered.

## Assessment of risk of bias

Risk of bias (RoB) assessments were performed using the Cochrane RoB for trials [21], with outcome specific domains (e.g., blinding of outcome assessors) assessed at the outcome level. A final judgement for each outcome was provided [22], which was used to inform the GRADE framework. ROBINS-I was used for observational studies of interventions [23]. Any studies that did not consider or adjust for the detection and quantification of fetal-maternal hemorrhage and the Rh status of the newborn, were considered as critical risk for the confounding bias question, as determined *a priori* (see protocol for details). RoB assessments were performed by one reviewer and verified by another (CH, LE, MT), with disagreements resolved through discussion.

## Data synthesis

Raw data were collected and displayed in summary tables, with details on timing and dosage of RhIG. Relative and absolute effects with 95% confidence intervals (CI) were calculated to

facilitate presentation of outcome data according to the GRADE summary of findings and evidence profile tables. Results from trials were analyzed separately from the results from the observational studies.

**Meta-analysis.**   Data was synthesised in Review Manager (version 5.3) [24], and results were presented in Forest Plots using a Peto Odds Ratios (OR), as there were small numbers of events [25]. Results were pooled regardless of the level of heterogeneity, however, this was noted in the results section when there was heterogeneity that was considered substantial ($I^2$ = 50–75%) or considerable ($I^2>75\%$), as guided by the Cochrane Handbook (section 9.5.2) [25]. When evaluating different dosages, as studies could contribute to more than one dosage subgroup, results are shown in Forest Plots, but a summary statistic was not estimated.

**Small study effects.**   An assessment has not been performed due to an insufficient number of studies [26], and has been reflected in the GRADE tables under 'other considerations' domain.

**Sensitivity analyses.**   As all studies were either moderate or high RoB (for trials) and critical risk (for observational studies), no sensitivity analyses were performed based on RoB.

**Rating the certainty of evidence.**   We assessed the certainty of the evidence using the GRADE approach [27], using GRADEpro GDT (https://gradepro.org/). We followed the GRADE guidance for determining the extent of the RoB for the body of evidence [28], and to evaluate imprecision [29]. The certainty of the evidence was rated by one reviewer and verified by a second. Discrepancies were resolved through discussion.

## Amendments to the protocol

We used the Cochrane RoB tool to evaluate quasi-randomised trials, rather than ROBINS-I, as the participants were not truly randomised (e.g., birth date). Although many outcomes were rated at high RoB, we felt it was still valuable to provide a meta-analysis, as there were notable effects for some outcomes. Lastly, due to low event rates, fixed effects models (due to the use of Peto odds ratios) were used instead of random-effects models.

## Results

### Study selection and characteristics

Database searches for RCTs yielded 530 citations. After removing duplicates and adding 24 records found in the supplemental searching, a total of 467 unique citations were evaluated based on the title and abstract, and 109 were further evaluated at full-text. Thirteen studies, representing 11 unique trials, were included (Fig 1). S4 File provides a list of the studies excluded at full-text by reason.

Some trials had companion papers providing additional results for a subgroup of low-risk women [30], or provided additional dosage levels [31]. Study characteristics are summarized in Table 2. Studies ranged from 14 to 4,865 women (median: 740), representing one to 43 sites (one study did not report the number of sites [32]). Trials were conducted in Argentina, Australia, Canada, France, Germany, the Netherlands, Scotland, the UK, and the USA. Most trials were performed in the 1960's and 1970's [30, 33–39], and several had multiple publications providing data at different time points or from different countries (footnoted in Table 2). Six trials evaluated postpartum RhIg [30, 33–35, 38, 40], three trials evaluated antenatal RhIg [32, 41, 42], two trials evaluated RhIg given to women who experienced abortion [37, 39], and two trials evaluated antenatal RhIg given either intramuscularly or intravenously [43, 44]. All but one trial [42] reported the women who were sensitized within the first year postpartum or at a subsequent Rh-positive pregnancy, and four studies reported adverse events (AEs) [39, 42–44].

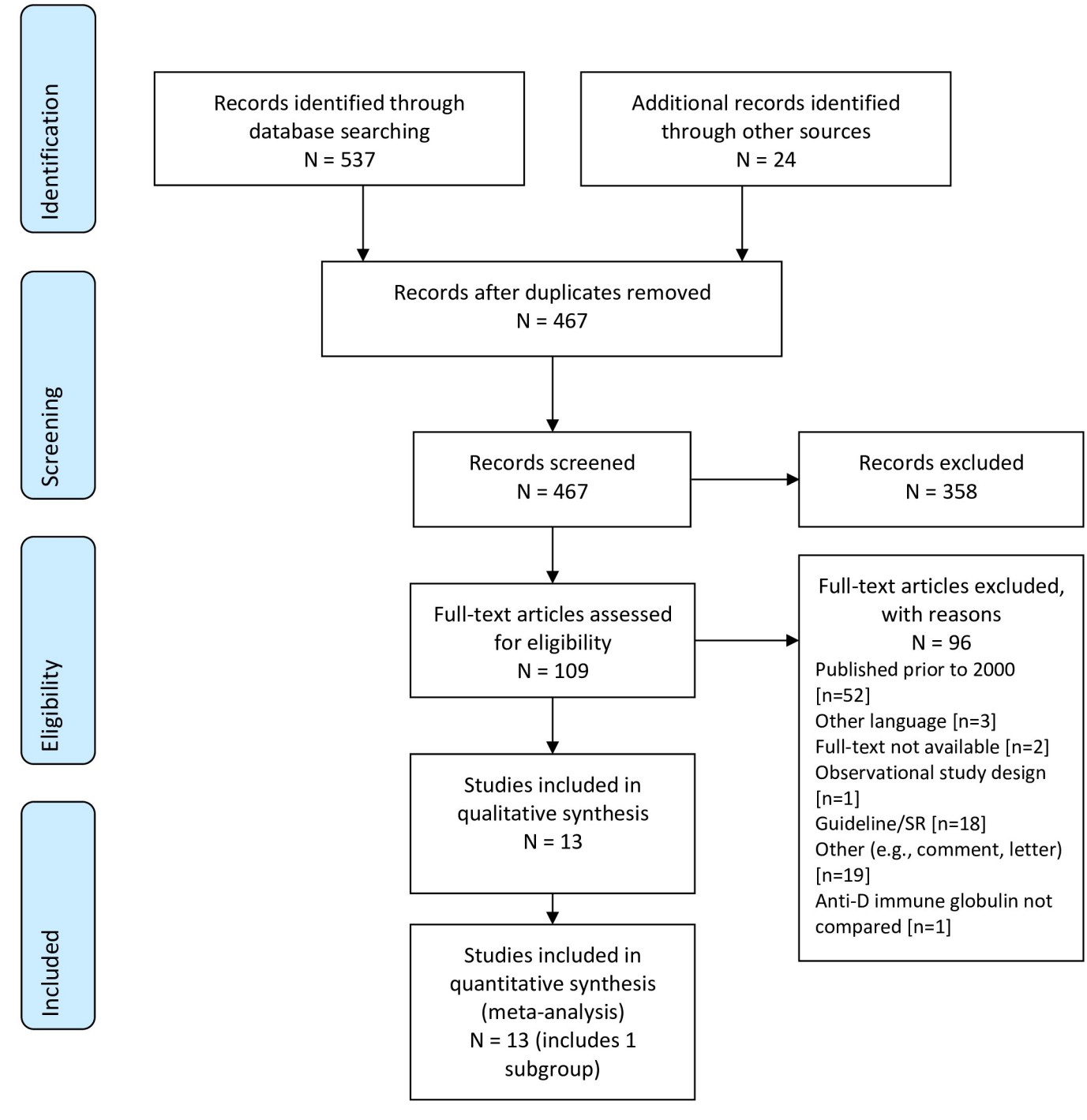

**Fig 1. PRIMSA flow diagram for RCTs.**

The database searches for comparative observational studies yielded 1099 citations. After removing duplicates and adding 17 records from the supplemental searching, 571 unique citations were evaluated based on the title and abstract, of which 101 were evaluated at full-text. Eight studies, representing seven cohorts, were included (Fig 2). S5 File provides a list of the studies excluded at full-text by reason.

**Table 2. Study characteristics of randomized controlled trials.**

| Author Year, Study design Funding | Setting & Follow-up | Participant demographics | Treatment groups: description of intervention (n providing results) | Outcomes |
|---|---|---|---|---|
| **Bichler 2003** [43], RCT | 7 gynecological practices in Germany | 14 Rh-negative women, aged ≥18 years, <28 weeks of gestation, with biological fathers who were Rh-positive | **Group 1**: Rhophylac (300 μg) given intramuscularly at 28-weeks gestation (n = 8) | • Antibodies |
| Funding: not reported | Follow-up: 6 months postpartum | **Age**: Range: 21–37 years | **Group 2**: Rhophylac (300) μg given intravenously at 28-weeks gestation (n = 6) | • Adverse events |
| | | **Race/ethnicity**: 100% Caucasian | Groups given RhIg within 72 hours after delivery of an RhD-positive child (route of administration up to the discretion of the investigator) | |
| **Chown 1969** [33], RCT[A] | 6 centres in Canada | 1716 non-immunised Rh-negative women with no Rh antibodies at delivery, with Rh-positive, ABO compatible infants | **Group 1**: RhIg (1.5 mL with approximately 435 μg of anti-D) given by intramuscular injection. Given within 72 hours after delivery (n = 852) | • Immunized |
| Funding: Department of National Health and Welfare, the National Institutes of Health | Follow-up: 6 months postpartum; during 2nd pregnancy | **Age**: not reported | **Group 2**: RhIg (0.5 mL with approximately 145 μg of anti-D) given by intramuscular injection. Given within 72 hours after delivery (n = 358) | |
| | | **Race/ethnicity**: not reported | **Group 3**: No treatment (n = 500) | |
| **Combined study 1971** [34], quasi-RCT (alternating)[B] | 4 centres in the UK and USA (Baltimore) | 349 Rh-negative primiparae women, free of antibodies, with maternal blood showing the presence of ≥0.2 mL of circulating fetal blood, who had just delivered ABO-compatible Rh-positive baby | **Intervention**: 5 ml of anti-D gammaglobulin [1000 microgram (England) and 1000 to 5000 micrograms (Baltimore) of IgG anti-D] given by intramuscular injection. Given within 36 hours of delivery (n = 173) | • Antibodies |
| Funding: Nuffield Foundation; Research Committee of the United Liverpool Hospitals | Follow-up: 6 months postpartum; during 2nd pregnancy | **Age**: not reported | **Comparator**: No treatment (n = 176) | |
| | | **Race/ethnicity**: not reported | | |
| **Dudok De Wit 1968** [35], quasi-RCT (odd-numbered birthdays) | 10 Specialty care clinic (e.g., Ob/gyn) in the Netherlands | 740 Rh-negative women who delivered a Rh-positive child, irrespective of ABO compatibility | **Intervention**: Anti-d Immunoglobulin (250 μg/ml) given by intramuscular injection. Given within 24 hours after delivery (n = 333) | • Antibodies |
| Funding: not reported | Follow-up: up to 6 months postpartum | **Age**: not reported | **Comparator**: no treatment (n = 329) | |
| | | **Race/ethnicity**: not reported | | |
| **Huchet 1987** [41], quasi-RCT (born in even or uneven years) | 23 maternity units in Paris, France | 1969 Rh-negative primiparous women | **Group 1**: RhIg (100 μg) given by intramuscular injection. Given at 28 and 34 weeks gestation (n = 599, among those with Rh-positive babies) | • Immunized |
| Funding: not reported | Follow-up: up to 12 months postpartum | **Age of participants**: not reported | **Group 2**: No treatment (n = 590, among those with Rh-positive babies) | |
| | | **Race/ethnicity**: not reported | In both groups, women with a Rh-positive baby were given postnatal RhIg (100 μg) | |
| **Lee 1995** [32], RCT | Maternity units (number not reported) in the UK | 2541 Rh-negative women in their first pregnancy recruited before 28 weeks gestation | **Group 1**: RhIg (250 iu/50 μg) (route of administration not reported), given at 28 and 34 weeks gestation (50μg RhIg at each administration) (n = 513, among those with Rh-positive babies) | • Anti-D |
| Funding: not reported | Follow-up: 6 months postpartum | **Age**: not reported | **Group 2**: No treatment (n = 595, among those with Rh-positive babies) | |
| | | **Race/ethnicity**: not reported | | |

(*Continued*)

**Table 2.** (*Continued*)

| Author Year, Study design Funding | Setting & Follow-up | Participant demographics | Treatment groups: description of intervention (n providing results) | Outcomes |
|---|---|---|---|---|
| **MacKenzie 2004** [44], RCT | 22 centres (17 UK, 5 USA) | 432 Rh-negative women without evidence of Rh(D) sensization, with known Rh(D) positive partners | **Group 1**: Rhophylac (300 μg) given intramuscularly (n = 216) | • Anti-D |
| Funding: not reported | Follow-up: 6 months postpartum | **Age**: Mean (SD): 29.1 (5.5) years; Range: 17.6–43.0 years | **Group 2**: Rhophylac (300 μg) given intravenously (n = 216) | • Adverse events: mild soreness, pain, mild itching, headache |
| | | **Race/ethnicity**: Caucasian: 400 (92.6%), Asian: 17 (3.9%), Afro-Caribbean: 7 (1.6%), Oriental: 0, Other: 8 (1.9%) | Both groups given at 28 weeks gestation and within 72 hours after delivery of a RhD-positive child. Additional doses of study drug were administered as required at the time of any potential sensitizing event or in response to an excessive FMH. | |
| **Medical Research Council 1974** [40], RCT | 9 hospitals in the UK | 2000 white Rh-negative primiparae married women whose serum had no detectable anti-D at the time of delivery, with no history of abortion or blood transfusion, with Rh-positive and ABO compatible infant | **Group 1**: RhIg (200μg/2ml) (n = 459) | • Anti-D presence (IAGT) |
| Funding: not reported (likely MRC) | Follow-up: 6 months postpartum; during 2nd pregnancy | **Age**: not reported | **Group 2**: RhIg (100ug/2ml) (n = 443) | |
| | | **Race/ethnicity**: 100% white | **Group 3**: RhIg (50ug/2ml) (n = 452) | |
| | | | **Group 4**: RhIg (20ug/2ml) (n = 446) | |
| | | | All groups were given RhIg by injection within 36 hours of delivery. | |
| **Stewart 1978** [39], RCT | Three outpatient facilities in Washington, DC, Chicago and Sacramento | 1027 Rh-negative pregnant women undergoing vacuum abortion at ≤12 weeks gestation (755 included at follow-up) | **Group 1**: MICRhoGAM (50 μg) given intramuscularly immediately following the abortion procedure (n = 691) | • Antibodies |
| Funding: not reported | Follow-up: 4 to 6 months following abortion | **Age**: 14–44 years, mean 22.5 years | **Group 2**: MICRhoGAM (300 μg (given intramuscularly immediately following the abortion procedure (n = 64) | • Adverse events |
| | | **Race/ethnicity**: 90% white | | |
| **Visscher 1972** [37], RCT | 3 hospitals in the USA | 57 $Rh_o$ (D) and $D_u$ negative women with $Rh_o$ (D) positive fathers | **Group 1**: RhIg (300 μg) given by intramuscular injection (n = 19) | • Sensitized |
| Funding: John A. Hartford Foundation Inc., New York City through the Blodgett Memorial Hospital Research Department | Follow-up: 6 months postpartum | **Age**: not reported | **Group 2**: Homologous gamma globulin (no demonstrable anti-Rho (D) antibody) 1 mL given by intramuscular injection (n = 29) | |
| | | **Race/ethnicity**: not reported | Group 1 and 2: Given within 72 hours after a spontaneous complete abortion or operative termination of a spontaneous incomplete abortion | |
| | | | **Other**: after randomization was broken, 9 additional participants were given no treatment | |

(*Continued*)

**Table 2.** (Continued)

| Author Year, Study design Funding | Setting & Follow-up | Participant demographics | Treatment groups: description of intervention (n providing results) | Outcomes |
|---|---|---|---|---|
| **White 1970** [38] & **Ascari 1968** [31], RCT[C] | 43 centres in Canada, Australia, Argentina, Scotland, United States, and Germany | 4,865 Rh-negative, previously unimmunized to the Rh antigen women, with a Rh positive, ABO compatible infant | **Group 1**: RhIg (Rho-GAM) [4.5 to 5 ml of RhIg containing 1000ug to 1200ug of anti-Rh per ml (4000–6000 ug total)] given intramuscularly. Given within 72 hours of delivery | • Antibodies |
| Funding: not reported | Follow-up: 6 months postpartum; during 2nd pregnancy | **Age**: not reported | **Group 2**: RhIg (Rho-GAM) [1 ml of RhIg containing no less than 300 ug anti-Rh] given intramuscularly. Given within 72 hours of delivery | |
| | | **Race/ethnicity**: not reported | **Group 3**: No treatment. Either an equivalent intramuscular injection of gammaglobulin solution devoid of anti-Rh or no injection at all (n = 1476) | |
| Note: dose subgroup data from Ascari 1968. | | | | |
| **White 2019** [42], | Western Australian tertiary maternity | 280 Rh-negative pregnant women attending antenatal care, aged ≥18 years, <30 weeks pregnant (277 included at follow-up) | **Group 1**: One dose RhIg (1500 IU) at 28 weeks of pregnancy given by intramuscular injection (n = 138) | • Adverse events |
| RCT | Hospital | **Age**: Mean (SD): 30.9 (5.0) single dose; 31.2 (5.0) two-dose | **Group 2**: Two doses RhIg (625 IU each) at 28 and 34 weeks of pregnancy given by intramuscular injection (n = 139) | |
| Funding: Women and Infants Research Foundation | Follow-up: at the time of delivery | **Race/ethnicity**: not reported | | |
| **Woodrow 1971** [30], quasi-RCT (alternating cases)[D] | 5 maternity units in the UK and USA | 715 women showing less than an estimated 0.2 ml of Rh-positive fetal blood in their circulation after delivery | **Group 1**: 1 ml of a gammaglobulin solution containing RhIg (200 µg) given intramuscularly. Given within 36 hours of delivery (delays of up to three to five days occasionally occurred) (n = 353) | • Antibodies |
| Funding: not reported | | **Age**: not reported | **Group 2**: No treatment (n = 362) | |
| | Follow-up: 6 months postpartum; during 2nd pregnancy | **Race/ethnicity**: not reported | | |

A **Western Canadian Trial**: Primary study: Chown 1969; Other publications not used: Buchanan 1969 [45], Godel 1968 [46]

B **Combined Study (England and Baltimore)**: Primary study: No author listed 1971 [34]; Other publications not used: Clarke 1965 [47], Clarke 1968 [48], Finn 1968 [49], Woodrow 1965 [50], No author listed 1966 [51]

C **International trial**: Primary study: White 1970 [38]; Subgroup data for dosage taken from Ascari 1968 [31]; Other publications not used: Ascari 1969 [52], Bishop 1968 [53], Bishop 1969 [54], Bryant 1969 [55], Freda 1966 [56], Freda 1967 [57], Jennings 1968 [58], Pollack 1968 [59], Robertson 1968 [60], Robertson 1969 [61], Stenchever 1970 [62], Symposium 1971 [63]

D Part of the Combined study, but reports only on those with <0.2 ml of Rh-positive fetal blood in their circulation. All other Combined study publications do not include these women.

Study characteristics have been summarized in Table 3. Three studies were conducted in the UK, two studies in Canada, and one each in Sweden and Denmark. Studies were published between 1978 and 2013. Study sizes ranged from 117 to 27,926 participants. One study each compared postpartum RhIg to no treatment [64], combined antenatal and postpartum RhIg to no treatment [64], two-dose antenatal plus postpartum treatment to one-dose antenatal plus postpartum treatment [4], different interval of RhIg administration between injection and delivery [65], treatment to no treatment after amniocentesis [66], and five studies (four cohorts) compared antenatal plus postpartum treatment to postpartum treatment only [66, 67–70]. No observational studies reported AEs.

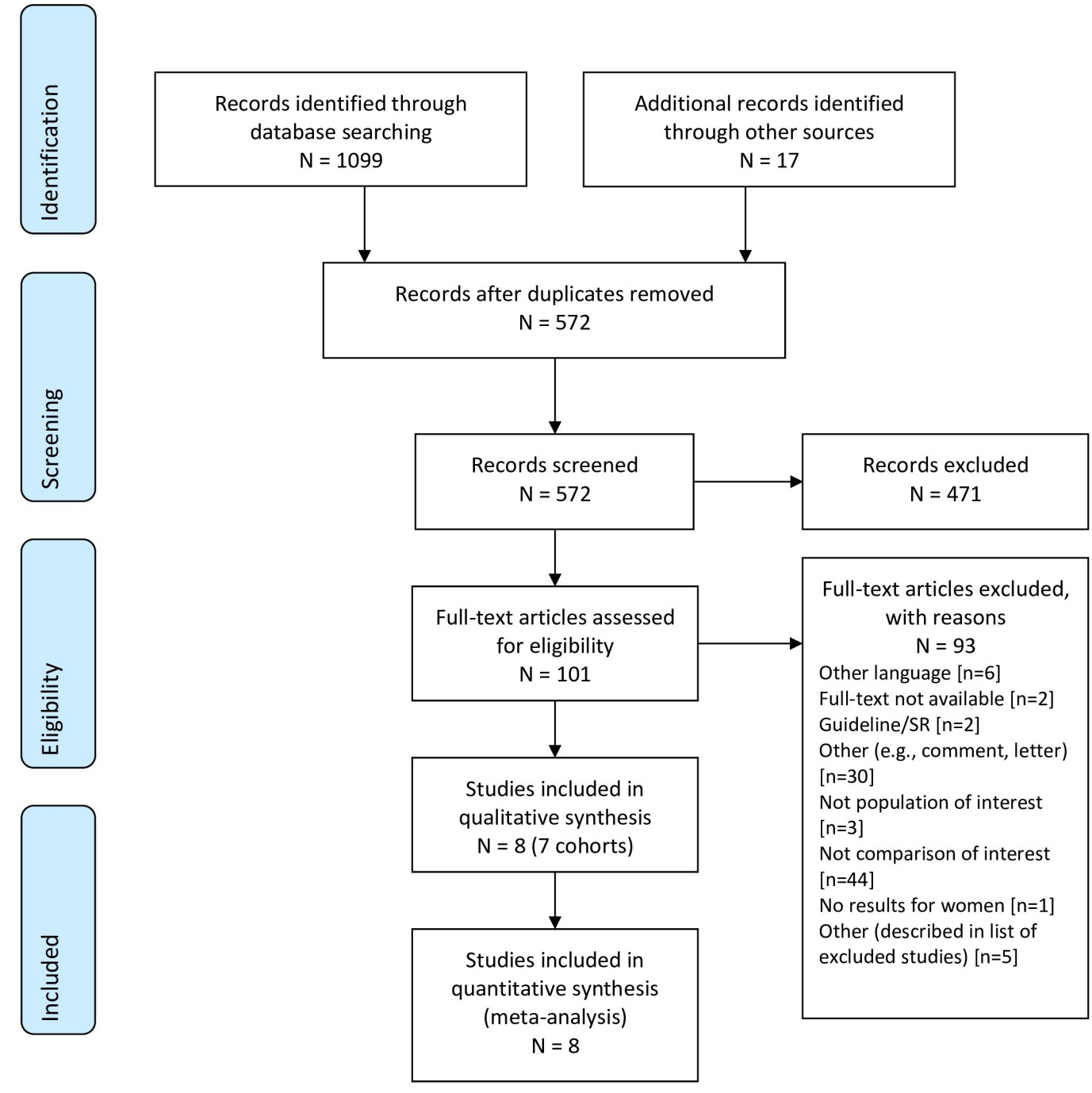

**Fig 2. PRIMSA flow diagram for observational studies.**

No RCTs reported on women who received RhIg after invasive fetal procedures. No RCTs or observational studies reported on women who received RhIg due to ectopic or molar pregnancy, maternal abdominal trauma, antepartum hemorrhage, or post-term RhIg (i.e., after 40 weeks' gestation).

**Table 3.  Study characteristics of observational studies.**

| Author Year Funding | Setting & Follow-up | Participant demographics | Treatment groups | Outcomes |
|---|---|---|---|---|
| **Bowman 1978** [4], Canada | Winnipeg hospitals | 1357 Rh-negative pregnant women who delivered Rh-positive babies. | **Group 1**: RhIg (300 µg) at 28 and 34 weeks (n = 1204) | • Rh isoimmunization |
| Funding: National Health grant; MRC of Canada grant | Follow-up: up to 6 months after delivery | | **Group 2**: RhIg (300 µg) at 28 or 34 weeks (n = 153) | |
| | | | All women were also given postnatal RhIg. | |
| **Bowman & Pollack 1978** [65], Canada | Primary care | **Group 1**: Women who had received antenatal and postnatal RhIg in all previous Rh-positive pregnancies and abortions. | **Group 1**: RhIg (300 µg) with interval between injection and delivery ≤8 weeks in most recent pregnancy | • Immunization |
| Funding: not reported | Follow-up: 6 months after delivery | **Group 2**: Rh negative multigravidas who had received RhIg only after delivery or not at all after previous Rh-positive pregnancies or abortions. Rh isoimmunization developing in women in this group may be due not to a failure of antenatal prophylaxis but to "sensibilization" as a result of inadequate treatment after previous pregnancies. | **Group 2**: RhIg (300 µg) with interval between injection and delivery >8 and < 16 weeks in most recent pregnancy | |
| **MacKenzie 1999** [64], UK | Two English counties (Oxfordshire and Northamptonshire) | Non-sensitized Rh-negative pregnant nulliparae | **Group 1** (Oxfordshire): RhIg (500 IU) at 28 and 34 weeks gestation given intramuscularly, plus standard prophylaxis postpartum (for all Rh-negative women regardless of baby status) and at other at risk occasions during the antenatal period (n = 3320) | • Rh sensitized second pregnancy |
| Funding: Bio Products Laboratories for financial assistance in conducting the retrospective analyses | Follow-up: second pregnancy | | **Group 2** (Northamptonshire): Antenatal prophylaxis was not offered, but those delivering a Rh-positive baby received RhIg (500 IU) (n = 3146) | |
| | | | **Group 3** (both counties 1980–1986): No RhIg at any time (n = 5971) | |
| **MRC by Working Party on Amniocentesis 1978** [66], UK | Hospitals in the UK | 133 Rh-negative amniocentesis women who had Rh-positive babies | **Group 1**: RhIg (dose not specified) following amniocentesis (n = 59) | • Immunized |
| Funding: not reported | Follow-up: not reported | Demographics not provided for this subgroup of women. | **Group 2**: No RhIg following amniocentesis (n = 58) | |
| **Tiblad 2013** [68], Sweden | Stockholm, Sweden (approximately 80 maternity care centres and six delivery units) | Prospective cohort: RhD negative pregnant women who had received RAADP and who delivered between January 1, 2010 and March 31st 2012. | **Group 1**: A single injection RhIg (250–300 µg) administered intramuscularly in pregnancy week 28–30 to Rh-negative women with a Rh-positive fetus. The study participants also received extra RhIg at events during pregnancy with increased risk for fetal-maternal hemorrhage (FMH) and after delivery (n = 4521 women participating with 4590 pregnancies) | • Immunization |
| Funding: Study funded by Stockholm County Council. The anti-D immunoglobulin used was sponsored by the pharmaceutical company Octapharma Nordic AB | Follow-up: during the study period for Group 1 and in the first trimester of a subsequent pregnancy for Group 2. | Reference cohort: RhD negative women giving birth in the same region between 2004–2008, prior to RAADP. | **Group 2**: Postnatal RhIg (250–300 µg) in non-immunised women was introduced in the early 1970s. RhIg prophylaxis has also been administered after interventions with risk of FMH (e.g., chorionic villus biopsies, amniocentesis, cordocentesis, external cephalic version) as well as after surgical terminations of pregnancy and spontaneous or induced abortion after 12 weeks of gestation. | |

*(Continued)*

**Table 3.** (Continued)

| Author Year Funding | Setting & Follow-up | Participant demographics | Treatment groups | Outcomes |
|---|---|---|---|---|
| **Thornton 1989** [70] & **Tovey 1983** [67], UK | 17 hospitals in West Yorkshire | 4069 Rh-negative pregnant women, 3115 gave birth to Rh-positive babies. | **Group 1** (1980–1981): RhIg (100 μg/ 500IU) intramuscularly given at 28 and 34 weeks (n = 2069 total, 1238 with Rh-positive babies) | • Immunization during first pregnancy |
| Funding: not reported | Follow-up: up to subsequent pregnancy | | **Group 2** (1978–1979): No antenatal RhIg (n = 2000, 1881 with Rh-positive babies) | • Immunization in subsequent pregnancy |
| | | | Both groups received RhIg (100 μg) after delivery if the Kleihauer count was normal. High fetal bleeds necessitated a higher dose (not specified) | |
| **Trolle 1989** [69], Denmark | Hospital in Kolding, Denmark | 700 Rh-negative women who delivered Rh-positive babies | **Group 1** (1980–1985): RhIg (300 μg) at 28 weeks. The day after delivery, if Kleihauer count was <15 ml blood and the baby was Rh-positive, the mother received RhIg (200 μg) (n = 354) | • Immunization 10 months after delivery or in next pregnancy |
| Funding: not reported | Follow-up: 10 months postpartum or in next pregnancy | | **Group 2** (1972–1977): Except for the prenatal injection and the antibody screen test at 28 week, they received the same treatment and examinations as group 1 (n = 346) | |

## Risk of bias of included studies

In RCTs, RoB was moderate and high for sensitization and high for AEs (Table 4). Four studies did not use truly random methods for randomization (e.g., birth date) [30, 34, 35, 41] and six studies did not provide sufficient information to judge the method of randomization [32, 33, 38, 39, 42, 43]. The method of allocation concealment was not reported in seven of the trials [32, 33, 38–40, 42, 44]. As sensitization was determined by a blood test, blinding of the participants and personnel would likely not impact the results, and risk was considered low. However, the four trials that reported AEs were not able to blind participants to treatment, and thus a judgement of high risk was assigned. Only one trial reported that the outcomes assessors were blind to treatment group [35], with the remaining trials not providing any information. As no trials reported working from an *a priori* developed protocol, selective reporting was rated as high, unless AEs were reported. Source of funding was often not reported resulting in an unclear judgement for the 'other' bias domain.

In all comparative cohort studies, RoB was judged as critical, as none clearly explained or adjusted for the detection and quantification of fetal-maternal hemorrhage. One study stated that a higher dose RhIg was provided if there was a 'high fetal bleed', but this was not elaborated [67]. Another study stated that if the Kleihauer count was <15 ml blood and the baby was Rh-positive, the mother received RhIg, however, there was no description on how this was quantified and it is unclear if this was a typographic error and should have read >15 ml [69].

## Rating the certainty of evidence

Overall, the certainty of the evidence in RCTs for all outcomes were low and very low (S6 File). This was mainly due to the serious and very serious RoB, few events and small sample sizes (i.e., imprecision). However, for a few outcomes (e.g., treatment compared with no treatment given at delivery) there were large treatment effects identified, and evidence included a large number of participants. Therefore, this information should be considered when evaluating the certainty of the evidence.

**Table 4. Risk of bias for RCTs.**

| Author Year (sorted by date of publication) | Sequence generation | Allocation concealment | Blinding of participants and personnel | Blinding of outcome assessment | Incomplete outcome | Selective reporting | Other bias | Summary judgement |
|---|---|---|---|---|---|---|---|---|
| **Outcome: Rh alloimmunization** | | | | | | | | |
| Bichler 2003 [43] | ? | − | + | ? | + | + | ? | High |
| Chown 1969 [33] | ? | ? | + | ? | + | − | + | Moderate |
| Combined Study 1971 [34] | − | + | + | ? | ? | − | + | High |
| Dudok De Wit 1968 [35] | − | − | + | + | ? | − | ? | High |
| Huchet 1987 [41] | − | − | + | ? | − | − | ? | High |
| Lee 1995 [32] | ? | ? | + | ? | − | − | ? | High |
| MacKenzie 2004 [44] | + | ? | + | ? | + | + | ? | Moderate |
| MRC 1974 [40] | + | ? | + | ? | + | − | ? | Moderate |
| Stewart 1978 [39] | ? | ? | ? | ? | − | + | ? | High |
| Visscher 1972 [37] | + | + | + | ? | ? | − | + | Moderate |
| White 1970 [38] | ? | ? | + | ? | ? | − | ? | Moderate |
| Woodrow 1971 [30] | − | − | + | ? | ? | − | ? | High |
| **Outcome: Adverse events** | | | | | | | | |
| MacKenzie 2004 [44] | + | ? | − | − | ? | + | ? | High |
| Bichler 2003 [43] | ? | − | − | − | ? | + | ? | High |
| Stewart 1978 [39] | ? | ? | ? | ? | − | + | ? | High |
| White 2019 [42] | ? | ? | − | − | ? | − | + | High |

In comparative cohort studies, all outcomes were very low certainty, mainly due to the extremely serious risk in the RoB domain (S6 File). Some outcomes were also considered serious for indirectness as the population included women who had Rh-negative infants [64, 68] and serious for imprecision due to the number of events and sample size [65, 66]. The outcomes under the antenatal plus postpartum RhIg compared to postpartum RhIg had fewer than 400 events, however, they were not down-rated as the total number of participants was greater than 2000 and they had a narrow absolute confidence interval that did not cross the threshold of no effect [29].

## Synthesis of results

**1. Postpartum administration.** *1.1 Postpartum RhIg vs. No RhIg*. Five trials compared RhIg (dose ranging from 200 to 6000 μg) given between 24 and 72 hours after delivery (n = 4756) to no RhIg (n = 2843) [30, 33–35, 38]. Four of these trials provided information by dose [30, 31, 34, 35]. In the "any dose" comparison, the study by Dudok De Wit et al. [35] has been reported as a subgroup, as they included women irrespective of ABO status, and it has been shown that ABO incompatibility between fetal erythrocytes and maternal serum partially

protects the mother against Rh immunization [38]. The Combined study provides two dosage levels of 1000 μg (UK) and 1000–5000 μg (Baltimore) [34] (S6 File: 1.1.1).

At 6-months postpartum, fewer women who received RhIg at delivery compared to no RhIg became sensitized in both women who had ABO compatible babies [70 fewer sensitized women per 1,000 (95%CI: 67 to 71 fewer); $I^2$ = 73%; very low certainty] and regardless of ABO status [39 fewer sensitized women per 1,000 (95%CI 22 to 46 fewer); very low certainty] (S6 File: 1.1.2, 1.1.3). Separating the results by dose resulted in favorable treatment effects regardless of dose when compared to no treatment (very low certainty) (S6 File: 1.1.2, 1.1.4).

Three trials followed women until their next Rh-positive pregnancy [30, 34, 38], with 130 fewer sensitized women per 1,000 who received any dose of RhIg in previous pregnancy compared to no RhIg (95%CI: 117 to 139 fewer; very low certainty) (S6 File: 1.1.2, 1.1.5). When evaluated per dose, in the 200 μg and 1000 μg groups compared to no RhIg, fewer women became sensitized (very low certainty) (S6 File: 1.1.2, 1.1.6).

One comparative cohort study evaluated women who received RhIg or not [64]. Compared to no RhIg, four fewer women per 1,000 were sensitized in a second pregnancy who had received postpartum RhIg in a prior pregnancy (95%CI: 0 to 7 fewer; very low certainty) (S6 File: 1.1.2, 1.1.7).

*1.2 Higher-dose postpartum RhIg compared to Lower-dose postpartum RhIg*

Two trials provided data to compare different doses of RhIg within 72 hours of childbirth (S6 File: 1.2.1) [31, 40]. At 6-months follow-up and at the end of a second D-positive pregnancy [40], fewer women who received a higher dose became sensitized, compared to a lower dose. However, based on the CIs, it is possible that there is no difference between groups or that more women given higher-dose RhIg became sensitized (low and very low certainty) (S6 File: 1.2.2, 1.2.3, 1.2.4).

**2. Routine antenatal administration.** *2.1 Antenatal RhIg (any dose) vs. No RhIg.* Two trials compared antenatal treatment with RhIg (two 50 μg or two 100 μg doses) given at 28 and 34 weeks to no treatment [32, 41]. Results were provided for all women, regardless of Rh status of the baby, and among those who delivered an Rh-positive baby (S6 File: 2.1.1). As women who deliver a Rh-negative baby have no risk of being sensitized, the results below are presented only among the women who delivered a Rh-positive baby.

At delivery, 6 fewer women per 1,000 in the RhIg compared to no RhIg became sensitized, but it is possible that there is no difference between groups or that more women given RhIg became sensitized (95%CI: 9 fewer to 1 more; very low certainty). At 2 to 12 months postpartum, 8 fewer women per 1,000 who received RhIg became sensitized compared to no RhIg (95%CI: 0 to 12 fewer; very low certainty). When combining at delivery and at follow-up, 9 fewer women per 1,000 (95%CI: 2 to 11 fewer; very low certainty) became sensitized in the RhIg group compared to no RhIg (S6 File: 2.1.2, 2.1.3).

*2.2 Antenatal plus postpartum RhIg vs No RhIg.* MacKenzie 1999 evaluated women who received ante- plus postpartum RhIg (n = 3320) to no RhIg (n = 5971) (S6 File: 2.2.1) [64]. Following women into their second pregnancy, 8 fewer women per 1,000 who received antenatal and postpartum RhIg were sensitized compared to no RhIg (95%CI: 5 to 10 fewer; very low certainty) (S6 File: 2.2.2, 2.2.3).

*2.3 Two-dose antenatal RhIg vs One-dose antenatal RhIg.* White 2019 [42] compared 138 women who received one-dose of RhIg (1500 IU) at 28 weeks of pregnancy to 139 women who received two doses of RhIg-D (625 IU/dose) at 28 and 34 weeks of pregnancy (S6 File: 2.3.1). Compliance was the primary outcome reported in this study. There were no major AEs in either group, however, authors stated that the greater injection volume (>5 mL) for the single dose group initially made it more painful than for the standard regimen. Therefore, a more concentrated product, delivering the same dose in a smaller volume (2 mL) was used.

*2.4 Two-dose antenatal plus postpartum RhIg vs One-dose antenatal plus postpartum RhIg.* In a comparative cohort study by Bowman 1978, one group of women who received two doses (300 μg) of RhIg at 28 and 34 weeks (n = 1204) were compared to another group who received one dose (300 μg) of RhIg-D at either 28 weeks or 34 weeks gestation (n = 153) [4]. All women were also given postpartum RhIg if they delivered a Rh-positive baby (S6 File: 2.4.1). No women in either group were sensitized at delivery or at six months postpartum (very low certainty), therefore it is unclear if one or two antenatal doses is favourable (S6 File: 2.4.2).

*2.5 Antenatal plus postpartum RhIg vs Postpartum RhIg.* Five studies, representing four cohorts, evaluated ante- and postpartum RhIg compared to postpartum RhIg, with results provided at different follow-up time (S6 File: 2.5.1) [64, 67–70].

Tovey 1983 [67] compared 1238 women who received 100 μg of RhIg at 28 and 34 weeks, and a postpartum dose to 1881 women who received only a postpartum injection. The postpartum dose for both groups was 100 μg if Kleihauer count was normal and a higher dose for "high fetal bleeds". At six months postpartum, 6 fewer women per 1,000 who received antenatal and postpartum RhIg compared to postpartum RhIg were sensitized (95%CI: 1 to 8 fewer; very low certainty) (S6 File: 2.5.2, 2.5.3).

Tiblad 2013 [68] and Trolle 1989 [69] reported on prospective cohorts of women who received antenatal and postpartum RhIg and compared them to historical cohorts of women who received only postpartum RhIg. The timing of the outcome for the prospective cohort was postpartum from the most recent pregnancy, and for the historical cohort, the subsequent pregnancy. Compared to postpartum treatment, 2 fewer women per 1,000 who received ante- and postpartum treatment were sensitized postpartum or in the second pregnancy (95%CI: 1 to 3 fewer; $I^2$ = 63%; very low certainty) (S6 File: 2.5.2, 2.5.3).

MacKenzie 1999 [64] and Thornton 1989 [70] compared 3924 women who received ante- and postpartum RhIg to 3728 women who received only postpartum RhIg. Compared to postpartum treatment, 6 fewer women per 1,000 who receive ante- and postpartum treatment were found to be sensitized in the second pregnancy (95%CI: 3 to 7 fewer; very low certainty) (S6 File: 2.5.2, 2.5.3).

*2.6 Shorter interval (≤8 weeks) vs Longer interval (>8 and <16 weeks) between treatment and delivery*

Bowman & Pollack 1978 compared a single injection of RhIg (300 μg) with the interval between injection and delivery at either ≤8 weeks or >8 weeks but <16 weeks in the most recent pregnancy [65]. Results were presented among two groups of women; those who had received ante- and postpartum RhIg in all previous Rh-positive pregnancies and abortions, and those who had received RhIg only after delivery or not at all after previous Rh-positive pregnancies or abortions (S6 File: 2.6.1). In both groups of women, fewer became sensitized with a shorter interval between injection and delivery, but it is possible that there is little to no difference between intervals or that more 'shorter-interval' women became sensitized (very lower certainty) (S6 File: 2.6.2, 2.6.3).

**3. Abortion.** *3.1 RhIg vs. No RhIg (placebo) given after spontaneous abortion.* One trial compared treatment of RhoGAM (300 μg) to placebo given to 48 women after experiencing a spontaneous abortion between 8 and 24 weeks gestation (S6 File: 3.1.1) [37]. At 6-months follow-up, no women in either group became sensitized (low certainty) (S6 File: 3.1.2).

*3.2 Higher-dose vs Reduced-dose RhIg following first trimester vacuum abortion.* Stewart 1978 randomised women to the standard dose (300 μg) RhIg or reduced-dose (50 μg) RhIg following first trimester vacuum abortion among 755 women (S6 File: 3.2.1) [39]. At four to six months follow-up, no women in either group became sensitized (very low certainty) (S6 File: 3.2.2). Only one woman reported an adverse event, although the physician felt it was probably attributable to the abortion procedure, rather than the drug.

**4. Amniocentesis.** *4.1 Treatment (any time) vs no treatment.* A report produced for the Medical Research Council by the Working Party on Amniocentesis in 1978 compared women who received amniocentesis prior to 20-weeks gestation, of which 59 women were given RhIg (dosage not specified) after the procedure and 58 women had not (S6 File: 4.1.1) [66]. Three women who did not receive RhIg became sensitized, an absolute effect of 45 fewer women per 1,000. However, it is possible that there is little to no difference between groups or that more women who were given RhIg became sensitized (95%CI: 51 fewer to 13 more; very low certainty) (S6 File: 4.1.2, 4.1.3).

**5. Intramuscular versus Intravenous administration.** *5.1. Antenatal RhIg given intramuscularly vs Intravenously.* Two trials randomized women to RhIg administered intramuscularly (IM) or intravenously (IV), given in the 28th week of pregnancy and again within 72 hours after delivering a RhD-positive infant (S6 File: 5.1.1) [43, 44]. At 6 to 9 months postpartum, only one woman tested positive in the IM group, but was no longer positive when tested at 11.5 months [44]. It is unclear which method of administration, if any, is favourable (S6 File: 5.1.2, 5.1.3). Few adverse events were reported, and those reported tended to be mild (e.g., pain at the injection site).

## Discussion

A search for RCTs and comparative observational studies to evaluate the effectiveness of treating Rh-negative women with RhIg during the antenatal period and/or at delivery to prevent Rh alloimmunization resulted in 21 studies (13 RCTs and 8 cohort studies), reporting on 12 different comparisons. Existing SRs [6–8] have included nine of the trials identified in this review, however, search dates are older, RoB assessment was incomplete in the review with the largest number of included studies, GRADE was only performed in one review on two trials, and none included comparative observational studies. Although there was little overlap in comparisons between trials and cohort studies, results from cohort studies provide additional treatment comparisons, some of which have been shown to be effective (e.g., antenatal plus postpartum vs postpartum alone). Overall, there are some comparisons which report a beneficial effect for Rh-negative women at risk of Rh alloimmunization, however, the evidence is primarily from small trials rated at high RoB and observational studies rated at critical RoB, resulting in low and very low certainty in the evidence for all outcomes. Although there seems to be large treatment effects for women who receive postpartum prophylaxis when compared to no treatment, we are unsure if the magnitude of these effects are overestimated.

Many countries now have routine antenatal and peripartum anti-D prophylaxis programs, including Canada, the United States, Finland, Sweden and the UK, among others. Rates of hemolytic disease have decreased substantially since the introduction of the prophylaxis programs, but this may also in part be due to lower birth rates [71]. For example, in 1960 the total births per woman in Canada was 3.81 births, decreasing to 1.74 in 1980 and 1.49 births in 2018 [72]. Fewer births mean less chance of Rh alloimmunization impacting future pregnancies. There is little consensus from international guideline groups on the dosage of administration [9]. For example, the ACOG recommends 300 μg at 28 weeks, the SOGC and RCOG recommend 300 μg at 28 weeks or 120 μg at 28 and 34 weeks, and RANZCOG recommends 125 μg at 28 and 34 weeks. Additionally, the dosage recommended after a potentially sensitizing event after 12-weeks ranges from 100 μg to 300 μg. The results from this review provides evidence that postpartum prophylaxis (in any dose) is more effective than no treatment; there is no difference between a two-dose (300 μg) antenatal plus postnatal prophylaxis and a one-dose (300 μg) antenatal plus postnatal prophylaxis; antenatal plus postnatal prophylaxis is more effective than postnatal prophylaxis alone; and there is no difference between intramuscular

and intravenous administration. Among women who experienced abortion, no women became sensitized, thereby not providing any evidence for a 300 μg dose, 50 μg dose, or placebo. In one comparative cohort study, there was no difference between women who were treated and those who were not, however, this is common treatment for women undergoing this procedure [73].

## Implications for future research

Although there were 21 included studies (13 RCTs and eight comparative cohort studies), this evidence based was spread out over 12 comparisons, resulting in sparse data, and when synthesis was possible, resulted in considerable heterogeneity at times. Few studies followed women to a subsequent Rh-positive pregnancy, and only four studies reported adverse events. Among those that reported AEs, it was unclear how these data were collected, and what little information was given was poorly reported. Future studies may benefit from the development of a core set of patient-important outcomes, as promoted by the Core Outcomes Measures in Effectiveness Trials (COMET) initiative [74–76].

Evidence from well-performed SRs form the backbone of evidence-based clinical practice guidelines offering several advantages: robust, transparent methodology highlighting methodological flaws and biases in the existing literature while identifying knowledge gaps and research priorities. Using GRADE also informs the certainty upon which recommendations from clinical practice guidelines are made. Previous SRs of prevention of Rh alloimmunization to date have been limited by study design of inclusion, incomplete risk of bias, and/or evaluating the certainty of the evidence. Nevertheless, there is evidence of effectiveness of this treatment from early trials and real-world experience in established prevention programs in developed countries. Our review has determined that the magnitude of these treatment effects remains uncertain and the optimal dosing strategy remains unknown. In addition, several subgroups of women who experience potentially sensitizing events during pregnancy (e.g., cordocentesis, ectopic or molar pregnancy, maternal abdominal trauma, post-term pregnancy) are not represented in RCTs and/or comparative observational studies, and there appears to be little ongoing research in this area. Although the Cochrane review by McBain *et al.* (2015) [7] reported two ongoing studies, a search for full-text publications did not identify related publications. Further, a search of clinicaltrials.gov resulted in no new trial registrations in this area.

Despite uncertainty of the evidence surrounding the optimal dosing and/or treatment strategies, prophylaxis programs have shown international success and will continue to reduce the burden of hemolytic disease in the developed world. Remaining unanswered questions would require further study. The costs of running RCTs to answer these and other treatment questions may be prohibitive however, and there may not be an appetite among the scientific and clinical community to mount these trials in light of the clinical success of prevention programs to date. In medicine, in recent years, the availability of data from large volumes of patients, accessed quickly through the use of automated, high speed processing of electronic medical records and clinical data banks has emerged as an additional and complementary strategy to answer clinical questions. Although attractive in its speed and ability to identify disease patterns and assess correlations between variables from different data sets, questions remain regarding standardization and validity of the "big data" analytics, as well as security, privacy and transparency concerns [77, 78]. Further studies on prevention of Rh alloimmunization with RhIg may benefit from this type of data. Authors of future observational studies should consult Strengthening the Reporting of Observational studies in Epidemiology (STROBE) for reporting guidelines of observational studies [79].

Going forward, further refinements in Rh immunoprophylaxis programs will undoubtedly occur, given the current capacity to non-invasively determine fetal Rh-D genotype on cell free DNA analysis of maternal blood with PCR technology. Such a strategy to antenatally test all Rh negative women to allow targeted administration of Rh immunoprophylaxis to only women carrying known Rh-positive fetuses has been adopted nationally in several countries with high sensitivity, specificity and negative and positive prediction rates [80, 81]. With ongoing cost-benefit analyses, other countries may adopt this approach or streamline implementation of fetal genotyping to mothers already immunized, particularly in the developed world [82–84].

Hemolytic disease of the fetus and newborn due to Rh incompatibility remains a significant contributor to perinatal mortality and morbidity in low and middle income countries, in particular those where medical infrastructure is rudimentary, costs and supply of RhIg are prohibitive, delivery systems are underdeveloped, and population and family sizes are larger. An international call to action has been issued to address this global concern and inequity [1]. To date, these populations are underrepresented in the current comparative literature. Considerations of resource implications, harms versus benefits and patient values and preferences remain essential components to establishing quality healthcare practices [85] and further research addressing these issues in Rh immunoprophylaxis is warranted in these subpopulations.

## Strengths and limitations

The strength of our work lies in the use of an *a priori* protocol, peer-reviewing the search strategy, an extensive grey literature and supplemental search to capture trials published prior to 2000, and updating from a previously published guideline to reduce duplication of effort and research waste. However, there are some limitations to this work. Although we included only English and French language published trials, nine records were excluded because they were published in other languages. Additionally, four articles could not be retrieved.

## Conclusion

This systematic review adds to the existing research by including comparative cohort studies, performs risk of bias on all domains and outcomes, and evaluates the certainty of the evidence using the GRADE framework. Evidence of low to very low certainty was established for all outcomes in RhIg effectiveness studies published to date due to inherent risk of bias, few events and small sample sizes, and some heterogeneity in outcome results. Despite these limitations, large treatment effects for postpartum administration of RhIg within 72 hours of delivery resulted in fewer sensitized women at 6-months postpartum follow-up and provided protection in the next Rh-positive pregnancy. Minimal postpartum dosage for prevention remains uncertain, with dose levels in the range of 200 to 6000 μg. The administration of antenatal prophylaxis as an adjunct to postpartum prophylaxis had an additional, albeit smaller, treatment effect. Cumulative doses of 200 to 300 μg (single or consecutive doses) have proven effective. Evidence for a preventative effect for spontaneous or induced abortion after 8 weeks is inconclusive due to zero events in either group in the included studies. Evidence to support RhIg following fetal diagnostic or therapeutic procedures is lacking or inconclusive from RCTs and comparative cohort studies. There is no evidence supporting improved effectiveness for IV over IM administration of RhIg.

In the developed world, prevention of Rh alloimmunization with RhIg has been a triumph of the 20th century. Several questions regarding optimal effectiveness remain, however. It is hoped that insights gleaned from this systematic review will help inform future or updated

clinical practice guidelines and set priorities for future work in the field. Uncertainties remain in the evidence, therefore to optimize transparency and trustworthiness in guidelines, development committees should be encouraged to reflect on the certainty of the evidence when developing recommendations affecting clinical care.

## Supporting information

**S1 File. Search strategies.**
(DOCX)

**S2 File. Additional search details.**
(DOCX)

**S3 File. Screening forms.**
(DOCX)

**S4 File. List of excluded studies by reason (trials).**
(DOCX)

**S5 File. List of excluded studies by reason (observational).**
(DOCX)

**S6 File. GRADE and Results tables.**
(DOCX)

**S1 Checklist. PRISMA checklist.**
(DOCX)

## Author Contributions

**Conceptualization:** Candyce Hamel, Karen Fung-Kee-Fung.

**Data curation:** Candyce Hamel, Lindsey Sikora.

**Formal analysis:** Candyce Hamel.

**Funding acquisition:** Candyce Hamel.

**Investigation:** Candyce Hamel, Leila Esmaeilisaraji, Micere Thuku, Alan Michaud.

**Methodology:** Candyce Hamel.

**Project administration:** Candyce Hamel.

**Resources:** Candyce Hamel.

**Supervision:** Candyce Hamel.

**Validation:** Candyce Hamel, Leila Esmaeilisaraji, Micere Thuku, Alan Michaud.

**Visualization:** Candyce Hamel, Karen Fung-Kee-Fung.

**Writing – original draft:** Candyce Hamel, Karen Fung-Kee-Fung.

**Writing – review & editing:** Candyce Hamel, Leila Esmaeilisaraji, Micere Thuku, Alan Michaud, Lindsey Sikora, Karen Fung-Kee-Fung.

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
