## [Decision Letter · Decision Letter 0]

19 Aug 2020

PONE-D-20-22184

Antenatal and Postpartum Prevention of Rh Alloimmunization: A Systematic Review and GRADE analysis

PLOS ONE

Dear Dr. Candyce Hamel,

Thank you for submitting your manuscript to PLOS ONE. After careful consideration, we feel that it has merit but does not fully meet PLOS ONE’s publication criteria as it currently stands. Therefore, we invite you to submit a revised version of the manuscript that addresses the points raised during the review process.

We look forward to receiving your revised manuscript.

Kind regards,

Georg M. Schmölzer

Academic Editor

PLOS ONE

Journal Requirements:

Reviewers' comments:

Reviewer's Responses to Questions

**Comments to the Author**

1. Is the manuscript technically sound, and do the data support the conclusions?

Reviewer #1: Yes

2. Has the statistical analysis been performed appropriately and rigorously? 

Reviewer #1: Yes

3. Have the authors made all data underlying the findings in their manuscript fully available?

Reviewer #1: Yes

4. Is the manuscript presented in an intelligible fashion and written in standard English?

Reviewer #1: Yes

5. Review Comments to the Author

Reviewer #1: Antenatal and Postpartum Prevention of Rh Alloimmunization: A Systematic Review

and GRADE analysis

Thank you for the opportunity to review this systematic review and GRADE analysis. The body of work is extensive and comprehensive, addressing an issue of enduring importance in obstetrics. The subtleties of timing and dosage of “Anti-D” during pregnancy are well expounded. The review provides an excellent overview of the evidence to inform clinical practice guidelines, albeit identifying why there is variation in these, given the heterogeneity of studies.

I have only extremely minor suggestions:

Line 40 “where” is “were”

Table 2: “Medical Research Council 1974,[41] RCT

Funding: not Reported”

Wouldn’t this be MRC-funded?

Table 3: add the word “be” to Bowman & Pollock, participant demographics, “… in this group may [be] due …”

Table 3: MRC by Working Party on Amniocentesis… participant demographics, replace “subjects” with “women”.

6. PLOS authors have the option to publish the peer review history of their article (what does this mean?). If published, this will include your full peer review and any attached files.

Reviewer #1: **Yes: **Christine East

---

## [Author Response · Author response to Decision Letter 0]

24 Aug 2020

Reviewer #1: Antenatal and Postpartum Prevention of Rh Alloimmunization: A Systematic Review and GRADE analysis

Thank you for the opportunity to review this systematic review and GRADE analysis. The body of work is extensive and comprehensive, addressing an issue of enduring importance in obstetrics. The subtleties of timing and dosage of “Anti-D” during pregnancy are well expounded. The review provides an excellent overview of the evidence to inform clinical practice guidelines, albeit identifying why there is variation in these, given the heterogeneity of studies.

Response: Thank you for your time in reviewing our manuscript and for providing feedback. We have made the changes as suggested below.

I have only extremely minor suggestions:

Line 40 “where” is “were”

Response: Thank you for catching this, it has been corrected.

Table 2: “Medical Research Council 1974,[41] RCT

Funding: not Reported”

Wouldn’t this be MRC-funded?

Response: Good point. I imagine this is likely the case, but it is not explicitly stated in the publication. I have kept not reported, but added (likely MRC) after.

Table 3: add the word “be” to Bowman & Pollock, participant demographics, “… in this group may [be] due …”

Response: Thank you, this has been corrected.

Table 3: MRC by Working Party on Amniocentesis… participant demographics, replace “subjects” with “women”.

Response: OK, this has been changed from subjects to women.

---

## [Editor Report · Decision Letter 1]

26 Aug 2020

Antenatal and Postpartum Prevention of Rh Alloimmunization: A Systematic Review and GRADE analysis

PONE-D-20-22184R1

Dear Dr. Candyce Hamel,

We’re pleased to inform you that your manuscript has been judged scientifically suitable for publication and will be formally accepted for publication once it meets all outstanding technical requirements.

Kind regards,

Georg M. Schmölzer

Academic Editor

PLOS ONE
---

## [Editor Report · Acceptance letter]

28 Aug 2020

PONE-D-20-22184R1 

Antenatal and Postpartum Prevention of Rh Alloimmunization: A Systematic Review and GRADE analysis 

Dear Dr. Hamel:

I'm pleased to inform you that your manuscript has been deemed suitable for publication in PLOS ONE. Congratulations! Your manuscript is now with our production department. 

Kind regards, 

on behalf of

Dr. Georg M. Schmölzer 

Academic Editor

PLOS ONE